# Social conditions and mental health during COVID-19 lockdown among people who do not identify with the man/woman binomial in Spain

**Constanza Jacques-Aviñó**[1,2]*, **Tomàs López-Jiménez**[1,2], **Laura Medina-Perucha**[1,2], **Jeroen de Bont**[1,2,3,4,5], **Anna Berenguera**[1,2,6]

**1** Fundació Institut Universitari per a la Recerca a l'Atenció Primària de Salut Jordi Gol i Gurina (IDIAPJGol), Barcelona, Spain, **2** Universitat Autònoma de Barcelona, Bellaterra (Cerdanyola del Vallès), Spain, **3** ISGlobal, Barcelona, Spain, **4** Spanish Consortium for Research on Epidemiology and Public Health (CIBERESP), Madrid, Spain, **5** Universitat Pompeu Fabra, Barcelona, Spain, **6** Departamentd'Infermeria, Universitat de Girona, Girona, Spain

* cjacques@idiapjgol.org, cjacques18@yahoo.es

**Data Availability Statement:** Data cannot be shared publicly because of ethical restrictions. The Ethical Committee does not allow us to share the

## Abstract

Evidence suggests that non-binary people have poorer mental and physical health outcomes, compared with people who identify within the gender binomial (man/woman). Research on the impact of the COVID-19 pandemic on mental health has been conducted worldwide in the last few months. It has however overlooked gender diversity. The aim of our study was to explore social and health-related factors associated with mental health (anxiety and depression) among people who do not identify with the man/woman binomial during COVID-19 lockdown in Spain. A cross-sectional study with online survey, aimed at the population residing in Spain during lockdown, was conducted. Data were collected between the 8th of April until the 28th of May 2020, the time period when lockdown was implemented in Spain. Mental health was measured using the Generalised Anxiety Disorder 7-item (GAD-7) scale for anxiety, and the Patient Health Questionnaire (PHQ-9) for depression. The survey included the question: Which sex do you identify with? The options "Man", "Woman", "Non-binary" and "I do not identify" were given. People who answered one of the last two options were selected for this study. Multivariate regression logistic models were constructed to evaluate the associations between sociodemographic, social and health-related factors, anxiety and depression. Out of the 7125 people who participated in the survey, 72 (1%) identified as non-binary or to not identify with another category. People who do not identify with the man/woman binomial (non-binary/I do not identify) presented high proportions of anxiety (41.7%) and depression (30.6%). Poorer mental health was associated with social-employment variables (e.g., not working before the pandemic) and health-related variables (e.g., poor or regular self-rated health). These findings suggest that social inequities, already experienced by non-binary communities before the pandemic, may deepen due to the COVID-19 pandemic.

data publicly as our data contain sensitive personal information and cannot be fully anonymized. Our study has been approved by the Research Ethics Committee of the Institut de Recerca en Atenció Primària Jordi Gol i Gurina (IDIAPJGol) (REC reference 20/063-PCV). Data are available from the Research Ethics Committee of the Institut de Recerca en Atenció Primària Jordi Gol i Gurina (IDIAPJGol) (contact via cei@idiapjgol.info) for researchers who meet the criteria for access to confidential data. For more information on data availability restrictions you can contact the ethics committee at cei@idiapjgol.info.

**Funding:** This work was supported by Spain's Ministry of Science and Innovation through the Carlos III Health Institute and European Union ERDF funds (European Regional Development Fund) through the Research Network in Preventive Activities and Health Promotion in Primary Care (redIAPP, RD16/0007/0001). The funders had no role in study design, data collection and analysis, decision to publish, or preparation of the manuscript.

**Competing interests:** The authors have declared that no competing interests exist.

## Introduction

Lockdown measures were implemented in Spain, in the context of a state of emergency, to control and prevent the COVID-19 pandemic. This strategy has meant for the population to remain at home, to reduce mobility and to keep physical distance from others. Research in Spain, the United Kingdom and Italy has highlighted the negative impact of lockdown on mental health, especially on women and the younger population [1–3]. For some people from the lesbian, gay, bisexual, trans, queer, intersexual, and other (LGBTQI+), lockdown has meant to live with people who were unaware of (or negatively viewed) their sexual orientation or gender identity. Some LGBTQI+ groups in Spain have made public the harm that their community has had to endure (either because of fear of retribution or the increase of discrimination by the people they live with) during lockdown [4, 5].

Based on a medical-legal system, the distinction between being a man and a woman seems simple, as being assigned to a sexual category within the binomial man/woman merely depends on observing the individuals' external genitalia [6]. However, this approach ignores gender diversity and how individuals may assign their own gender identity (e.g., non-binary people) [7, 8]. The term "non-binary" could be understood as a category that includes a) people who do not identify within or in-between the man/woman classification; b) people who could experience man/woman identities separately; and c) people who do not identify their gender identity or who reject to identify with specific gender identities [9]. Non-binarism could include individuals who identify as trans and *queer* [10, 11]. However, not all non-binary people identify as trans and some trans people may identify within the man/woman binomial category [12]. Besides, there could be people who question the sex-gender dichotomies but do not identify with, or are unaware of, the term "non-binary".

Available evidence shows how discrimination is higher among non-binary people, compared to binary people, given the social and hegemonic rejection of non-binarism. Non-binary people also experience distress as they do not feel socially represented and visible [9]. According with this evidence, a previous study including cisgender, trans and non-binary people in Spain found that non-binary people received less social support from family and friends. They also experienced higher social isolation and cyberbullying rates, compared to other groups [13]. Besides, non-binary individuals have been reported to have higher prevalence rates of depression, anxiety and illicit drug use, in comparison with binary people [11].

Despite the increasing emergence of scientific and academic debates around the diversity in sexual-gender identities, the evidence on how sexual-gender identity may be linked to health outcomes is still scarce. Existing studies often have as their main focus on sexually transmitted infections. National health surveys in Spain are still embedded within the man/woman binominal, neglecting gender identity [14]. The same is true for electronic health records, which only allow classifying individuals as men or women, suggesting a lack of knowledge on sexual and gender diversity populations [15]. Based on these arguments, and assuming the theoretical and methodological complexity around the sex-gender system, this study aims to explore social and health-related factors associated with mental health during COVID-19 lockdown among people who do not identify with the man/woman binomial in Spain.

## Methods

A cross-sectional study was conducted among the general population (≥18 years) living in Spain during COVID-19 lockdown. Data were collected through an online survey between the 8th of April and the 28th of May 2020. Study data were collected and managed using REDCap (Research Electronic Data Capture) electronic data capture tools hosted at Fundació Institut Universitari per a la recerca a l'Atenció Primària de Salut Jordi Gol i Gurina (IDIAPJGol).

REDCap is a secure, web-based software platform designed to support data capture for research studies, providing (1) an intuitive interface for validated data capture; (2) audit trails for tracking data manipulation and export procedures; (3) automated export procedures for seamless data downloads to common statistical packages and (4) procedures for data integration and interoperability with external sources [16, 17]. Participants were recruited through online platforms and social networks using convenience and snowball sampling techniques through different institutions and personal accounts. The questionnaire was created by a team researcher including psychologists, statisticians and epidemiologists. We analysed if there were any incoherent answers and did not find any. The survey was piloted with people with different sociodemographic characteristics, making some changes in the survey so that it was easier to complete it. The average time to answer the survey was 10 minutes; which was clarified on the front page of the survey. Data collection was finalised when lockdown de-escalation started in Spain. This study has been informed by social determinants' and gender-based framework [18]. All members of the research team are highly sensitive towards the different axes of social inequities, and the multiple power structures that impact the population unequally.

The sample for this study was composed by people who replied "Non-binary" or "I do not identify" in the "What sex do you identify with?" survey question. People who selected the "woman" or "man" options, and those who did not reply, were excluded. Categories were excluding and participants could not tick more than one option. Data on the social impact among people who identified as "man" or "woman" are presented in another study [3]. For this article, the term sexual identity will be used and gender identity indistinctly. This term is understood as part of a self-belief system or how we self-identify and/or publicly identify. For the analyses, the sex-gender system is intrinsic to the term identity. We assume that respondents ticked the response categories with which they identified.

The main variable was "mental health". It was measured through the Generalised Anxiety Disorder 7-item (GAD-7) scale for anxiety, and the Patient Health Questionnaire (PHQ-9) for depression. Anxiety was defined as excessive worry and persistent restlessness related to different elements such as personal health, employment, social interactions and everyday life situations [19]. The anxiety variable was categorized as normal, mild, moderate and severe. Anxiety was considered as moderate and severe level. Depression was defined as feeling sadness, frustration or irritability frequently or most of the time [20]. The depression variable was categorized in none-minimal, mild, moderate and moderately-severe/severe. -Depression was considered as moderate and moderately-severe/severe level.

Several social and health-related variables were considered as independent variables. Social-related factors were: employment status before and after lockdown, housing and living conditions, violence at home and support from neighbours. Health-related factors were: experiences of COVID-19 (had COVID-19 diagnostic or symptoms, dead of loved one, fear of COVID-19 infection, COVID-19 is a problem for your economy), self-rated health and health-oriented behaviours (tobacco and alcohol consumption and physical activity). In addition, socio-demographics variables were: age, educational level, country of birth and working in essential work. To compare non-binary people with binary people, we did a matched analysis by choosing 4 binary people for each non-binary person. Their characteristics were as follows: 2 men and 2 women, same age range (±1 year) and same level of education.

Variables' frequencies and percentages were calculated in relation to the variables "Non-binary/I do not identify". Multivariate regression models were constructed to evaluate the association between mental health variables and socio-demographics and independent variables (social and health-related factors). In the non-binary database, there was only one missing value in educational level. Multiple imputation by chained equations with 10 imputed

datasets was applied to this value. We found that 90% of the imputations in this case were "University", so we then performed single imputation, assigning "University" to this missing value. Finally, the selected variables age, educational level, and other variables that were significant in the depression and anxiety, were included in the final model. Co-variables in the model were the same in both models to allow comparisons. The principle of parsimony was followed given the small sample size. Adjusted odds ratios (aOR) were calculated, as well as their corresponding 95% confidence intervals (IC95%). Analyses were performed using Stata 15.1.

### Ethical considerations

Ethical approvals were granted by the *Institut Universitari d'Investigació en Atenció Primària* IDIAPJGol Ethics Committee (20/063-PCV). Respondents' confidentiality was ensured. The submission of the answered questionnaire was considered to be their consent to participate in the study.

## Results

A total of 72 people reported to identify as non-binary (N = 35) or as to not identify by any other category (N = 37). This was equivalent to a 1% of the survey's sample (N = 7125). This study's main sample characteristics are available in Table 1. In this study, 41.7% presented anxiety symptoms and 30.6% of depression. The average age was 40.6 years (standard deviation: 13.2), 67.6% had completed university studies and 88.9% were born in Spain. A 27.8% were not working before lockdown, 19.4% were essential workers (those working during the state of emergency), and 51.4% reported worsened employment situations during lockdown. Regarding housing conditions, 13.9% declared that their house was not suitable to live in during lockdown. 28.2% reported being worried about the relationships with the people they lived with (living conditions) and 8.5% reported experiences of violence during lockdown (physical, economic, psychological and/or sexual).

### Anxiety

There was an association between anxiety and being unemployed before lockdown (aOR: 4.91. IC95%:1.13–21.26). Also, between anxiety and worsened employment status during lockdown (aOR: 4.99 IC95%:1.38–18.0-). There were no other statistically significant associations. See Table 2 for more details.

### Depression

Risk of depression was higher among the younger population group (aged 18–35) (aOR: 16.27, IC95%: 2.97–93.54), in people who were not working before lockdown (aOR: 36.11 IC95%: 4.01–296), among those whose employment status worsened during lockdown (aOR: 6.28 IC95%: 1.25–31.65) and in people with a regular or poor self-rated health (aOR: 13.58 IC95%: 1.15–160.0). There was an association between depression and fear about COVID-19 infection (aOR: 8.93 IC95%: 1.41–56.68), and the increase in alcohol consumption (aOR: 19.52 IC95%: 2.30–166).

When comparing our findings with binary people we observed that non-binary people presented more anxiety symptoms than binary people (41.1% vs 27.1%). While depressive symptoms were similar to binary people (30.6% vs 28.8%). In addition, we performed a stratified analysis between men and women (S1 Table). Our results showed that non-binary people had more anxiety than men (16.0%) and that this was statistically significant. They also had more

**Table 1. Mental health, sociodemographic characteristics and social and health-related factors among people who do not identify within the man/woman binomial during COVID-19 lockdown in Spain.**

| | Non-binary (n = 35) | I do not identify (n = 37) | Total (n = 72) |
|---|---|---|---|
| **GAD7** | | | |
| Normal/Mild | 20 (57.1%) | 22 (59.5%) | 42 (58.3%) |
| Moderate/Severe | 15 (42.9%) | 15 (40.5%) | 30 (41.7%) |
| **PHQ-9** | | | |
| None-minimal/Mild | 23 (65.7%) | 27 (73.0%) | 50 (69.4%) |
| Moderate/Moderately severe/Severe | 12 (34.3%) | 10 (27.0%) | 22 (30.6%) |
| **Age** | | | |
| 18–35 years | 14 (40.0%) | 12 (32.4%) | 26 (36.1%) |
| >35 years | 21 (60.0%) | 25 (67.6%) | 46 (63.9%) |
| **Educational level** | | | |
| Primary/Secondary | 15 (42.9%) | 8 (21.6%) | 23 (31.9%) |
| University | 20 (57.1%) | 29 (78.4%) | 49 (68.1%) |
| **Country of Birth** | | | |
| Spain | 31 (88.6%) | 33 (89.2%) | 64 (88.9%) |
| Other countries | 4 (11.4%) | 4 (10.8%) | 8 (11.1%) |
| **Employment status before lockdown** | | | |
| Working | 24 (68.6%) | 28 (75.7%) | 52 (72.2%) |
| Not working | 11 (31.4%) | 9 (24.3%) | 20 (27.8%) |
| **Essential work** | | | |
| No | 30 (85.7%) | 28 (75.7%) | 58 (80.6%) |
| Yes | 5 (14.3%) | 9 (24.3%) | 14 (19.4%) |
| **Employment condition** | | | |
| No change/Improved | 14 (40.0%) | 21 (56.8%) | 35 (48.6%) |
| Worsened | 21 (60.0%) | 16 (43.2%) | 37 (51.4%) |
| **Living conditions** | | | |
| Alone | 5 (14.3%) | 10 (27.0%) | 15 (20.8%) |
| Not alone | 30 (85.7%) | 27 (73.0%) | 57 (79.2%) |
| **Adequate housing conditions** | | | |
| No | 4 (11.4%) | 6 (16.2%) | 10 (13.9%) |
| Yes | 31 (88.6%) | 31 (83.8%) | 62 (86.1%) |
| **Concern relationships with people live with** | | | |
| No | 22 (62.9%) | 29 (80.6%) | 51 (71.8%) |
| Yes | 13 (37.1%) | 7 (19.4%) | 20 (28.2%) |
| **Violence at home** | | | |
| No | 30 (85.7%) | 35 (97.2%) | 65 (91.5%) |
| Yes | 5 (14.3%) | 1 (2.8%) | 6 (8.5%) |
| **Self-rated health** | | | |
| Good/Very good/Excellent | 31 (88.6%) | 32 (86.5%) | 63 (87.5%) |
| Regular/Poor | 4 (11.4%) | 5 (13.5%) | 9 (12.5%) |
| **COVID-19 diagnostic or symptoms** | | | |
| No | 29 (82.9%) | 34 (91.9%) | 63 (87.5%) |
| Yes | 6 (17.1%) | 3 (8.1%) | 9 (12.5%) |
| **Dead of loved ones** | | | |
| No | 29 (82.9%) | 29 (78.4%) | 58 (80.6%) |
| Yes | 6 (17.1%) | 8 (21.6%) | 14 (19.4%) |
| **Support from neighbours** | | | |

(*Continued*)

**Table 1.** (Continued)

| | Non-binary (n = 35) | I do not identify (n = 37) | Total (n = 72) |
|---|---|---|---|
| No | 9 (27.3%) | 7 (18.9%) | 16 (22.9%) |
| Yes | 24 (72.7%) | 30 (81.1%) | 54 (77.1%) |
| **Fear of COVID-19 infection** | | | |
| No | 11 (31.4%) | 21 (56.8%) | 32 (44.4%) |
| Yes | 24 (68.6%) | 16 (43.2%) | 40 (55.6%) |
| **COVID-19 is a problem for your economy** | | | |
| No | 5 (14.3%) | 8 (21.6%) | 13 (18.1%) |
| Yes | 30 (85.7%) | 29 (78.4%) | 59 (81.9%) |
| **Tobacco consumption** | | | |
| No use/Same use | 30 (85.7%) | 32 (86.5%) | 62 (86.1%) |
| Increased use | 3 (8.6%) | 5 (13.5%) | 8 (11.1%) |
| Decreased use | 2 (5.7%) | 0 (0%) | 2 (2.8%) |
| **Alcohol consumption** | | | |
| No use/Same use | 23 (65.7%) | 20 (54.1%) | 43 (59.7%) |
| Increased use | 5 (14.3%) | 9 (24.3%) | 14 (19.4%) |
| Decreased use | 7 (20.0%) | 8 (21.6%) | 15 (20.8%) |
| **Practice physical activity** | | | |
| No practice/Same practice | 16 (45.7%) | 17 (45.9%) | 33 (45.8%) |
| Increased practice | 7 (20.0%) | 3 (8.1%) | 10 (13.9%) |
| Decreased practice | 12 (34.3%) | 17 (45.9%) | 29 (40.3%) |

[1] GAD 7: Generalised Anxiety Disorder 7-item scale [2] PHQ-9: Patient Health Questionnaire

anxiety symptoms than women (38.2%), but this was not statistically significant. As for depression, women reported more depression-related symptoms (38.9%) compared to non-binary people. However, this difference was not statistically significant. In men, they presented 18.8% of depressive symptoms, reaching almost statistical significance (p-valor: 0.051) (S2 Table). In addition, when we introduced the non-binary people as an independent variable, we could observe that identifying a non-binary person was a risk factor for anxiety (ORa: 1.89 CI: 1.05–3.39) but not for depression (ORa: 0.93; CI: 0.50–1.74) (S3 Table).

Finally, in order to go into the different items in more detail of mental health we only found significant differences between binary and non-binary people in the item being so restless that it is hard to sit still of the GAD-7, while in the PHQ-9 there were no differences (S4 and S5 Tables). However, we differentiate between binary people. We found worse results among non-binary people compared to men on the following GAD-7 items: worrying too much about different things; having trouble relaxing and being so restless that it is hard to sit still. On the other hand, we found worse results among non-binary people compared to men on the following items of the PHQ-9: feeling down, depressed, or hopeless; trouble falling or staying asleep, or sleeping too much and feeling tired or having little energy (S6 and S7 Tables).

## Discussion

This study presents an overview of the social conditions and psychological impact during COVID-19 lockdown in Spain among a group of non-binary people or people who do not identify with the man/woman binomial category. Poorer mental health was associated with social-employment and health-related variables. Results highlighted the impact of employment

**Table 2. Associations between sociodemographic characteristics, social and health-related factors, and mental health in people who do not identify within the man/woman binomial during COVID-19 lockdown in Spain (N = 72).**

| | GAD7 | | PHQ-9 | |
|---|---|---|---|---|
| | aOR[3] (95%CI) | P-value[4] | aOR (95%CI) | P-value |
| **Age** | | | | |
| >35 years | 1.00 | | 1.00 | |
| 18–35 age | 2.19 (0.64–7.44) | 0.209 | 16.67 (2.97–93.54) | 0.001 |
| **Educational level** | | | | |
| Primary/Secondary | 1.00 | | 1.00 | |
| University | 0.71 (0.21–2.47) | 0.595 | 3.35 (0.62–18.27) | 0.162 |
| **Employment status before lockdown** | | | | |
| Working | 1.00 | | 1.00 | |
| Not working | 4.91 (1.13–21.26) | 0.033 | 36.11 (4.41–296) | 0.001 |
| **Employment condition** | | | | |
| No change/Improved | 1.00 | | 1.00 | |
| Worsened | 4.99 (1.38–18.0) | 0.014 | 6.28 (1.25–31.65) | 0.026 |
| **Self-rated health** | | | | |
| Good/Very good/Excellent | 1.00 | | 1.00 | |
| Regular/Poor | 3.10 (0.49–19.42) | 0.227 | 13.58 (1.15–160) | 0.038 |
| **Fear of COVID-19 infection** | | | | |
| No | 1.00 | | 1.00 | |
| Yes | 3.29 (0.95–11.33) | 0.059 | 8.93 (1.41–56.68) | 0.020 |
| **Alcohol consumption** | | | | |
| No use/Same use | 1.00 | 0.212 | 1.00 | 0.017 |
| Increased use | 2.14 (0.43–10.76) | | 19.52 (2.30–166) | |
| Decreased use | 3.67 (0.83–16.27) | | 6.64 (0.94–47.12) | |

Models were adjusted for age, education level, pre-lockdown employment status, employment condition as consequence of lockdown, self-rated health, fear of being infected with COVID-19 and changes in alcohol consumption during lockdown.

[1] GAD 7: Generalised Anxiety Disorder 7-item scale

[2] PHQ-9: Patient Health Questionnaire

[3] aOR: adjusted ordinal odds ratio.

[4] P-value: Statistical significance derived from using Wald test

status-related factors (before and during lockdown) and reporting poor self-rated health. Over 40% of the sample presented anxiety and one third reported depression, especially affecting those who are younger. These findings differ from findings from the same survey in people who identified as women or men. According to these data, 31.2% women and 17.7% men reported anxiety. Depression was reported in 28.5% of women and 16.7% of men [3]. To further explore our findings, we carried out a comparison. We found statistically differences in anxiety levels between non-binary and binary people. When stratified by gender, we observed that non-binary people and people identified as women had similar anxiety levels, but we observed statically significant lower levels of anxiety in men. This suggests that having more anxiety during lockdown makes non-binary people more similar to women. According to our previous findings, social factors such as lack of economic privilege and gender could influence poorer mental health outcomes [3].

It is possible that poor mental health among surveyed people was already experienced before lockdown, particularly for people who were unemployed prior to the pandemic and that had seen their socio-economic situation negatively affected. There was also an association

between poor mental health and worsening employment status during lockdown. This is consistent with other studies that have established a relationship between unemployment and poor mental health among the general population, related to a feeling of having fewer opportunities to find stable employment in the future [21]. Also, a study conducted with trans (feminine and masculine) and non-binary people showed that employment status was a significant predictor of health. In the reported study, those who were not working had higher probabilities of having a regular or poor health, frequent days experiencing physical and mental health problems, and having two or more chronic diseases, in comparison with employed people [22]. This could be explained due to the discrimination that non-binary people experience when seeking employment [13].

In our study, there was an association between poor self-rated health and depression. There is a vast amount of evidence on self-rated health as a predictor of morbidity and mortality and its association with social inequities [23]. Other studies including non-binary people and those who fluctuate between a binary and non-binary identity, have reported high rates of depression, compared to binary people. This has been attributed to a lack of social acceptance for the diversity in sex-gender identities [11, 12]. It could be suggested that depression was experienced before lockdown, given the structural nature of power hierarchies that maintain discriminatory practices towards people with non-normative identities [24]. In relation to this, the association between depression and the increase in alcohol consumption during lockdown could be explained as alcohol may be used as a strategy to temporally disconnect from negative feelings and thoughts. This suggests challenges to train primary healthcare and public health professionals. On one hand, because people with non-normative identities encounter barriers to access healthcare systems [24]. On the other hand, because identifying problematic alcohol use is challenging for health professionals, as there is a lack of sensitivity towards social determinants of health in people who do not identify with the binomial man/woman category [12]. This leads to question approaches that consider the individual as pathologic. Instead, a framework that considers the analysis of ideologies (institutional and scientific) that maintain and reproduce inequities should be taken [24]. Regarding COVID-19 risk perception, we found an association between fear of infection and depression. In previous studies, and in our previous work, this association was only found among women [3, 25].

Perceiving inadequate housing conditions during lockdown was not an adjustment variable for anxiety and depression. Nor was an adjustment variable concern about relationships with people participants were living with. However, it is important to note that previous studies have shown an association between housing problems and poor self-rated health [26]. Besides, non-binary individuals have been found to be at a higher risk for homelessness and living in unstable housing, compared to binary individuals [12]. It is then essential to consider housing conditions as a fundamental axis of health when future lockdown measures are implemented. Both structural housing elements and comfort should be considered. As for the reports of violence, it was not an explanatory variable for poor mental health. However, we would like to highlight that 8.5% of people who did not identify with the man/woman binomial declared having experienced violence during lockdown. This is a significantly higher proportion if compared with reports of violence by men (1.3%) and women (2.3%) in the same study [3].

There are limitations to consider. Data related to sexual orientation or sexual desire were not collected. Thus, this study cannot be framed within the LGBTQI+ population. Regarding the selected sample, we are unaware of how surveyed people interpreted the options "Non-binary" and "I do not identify". We also do not know the experience related to their sex at birth and how they have been treated by their social environment. Therefore, we make suggestions. Questions for a future survey could be: What was your sex at birth? What is your gender identity? And, what is your sexual orientation? Regarding the response categories, the research

team believes that it is relevant to include the option: I prefer not to answer. Also, the findings presented in this paper will be combined with a qualitative study. Participants for the qualitative study will be recruited through contact details that people taking part in the survey provided (with consent) to the research team.

Moreover, the survey did not have similar questions in different parts to assess inconsistency, nor could we limit it to be answered more than once. For the purposes of this study, we conducted a review of entry survey to assess internal consistency, for example, comparing age and education level. Another limitation is the small simple size, which meant that some estimations were statistically imprecise. Findings should then be interpreted with caution. On the other side, the small simple size did not allow to perform an intersectional analysis. This could have allowed to identify other axes of social inequities in our sample. However, the proportion of individuals that did not identify with the man/woman binomial is consistent with another study conducted in Spain with young people, which identified 70 individuals as non-binary [13]. Besides, given the study design, we are unaware of the situation that participants were in before lockdown. There are also strengths in our study, as it allows reflecting on the social conditions that people who did not identify with the man/woman binomial lived in during lockdown. These are relevant findings if we consider the high number of publications available on the COVID-19 pandemic, in which gender diversity is overlooked. In addition, this study contributes to the debate on the categorisation of sex-gender identities in health surveys and other registries, to question and challenge normative ways of classification.

People's embodiment, as a biological (and health) construct, is based on a repository of experiences that are part of social processes [6, 7, 27]. Questioning biological essentialism is thus key to this discussion. The issues of biological accounts of sex-gender, and the need for the reassessment of gender analyses are crucial [28]. Both sex and gender should not be considered unidimensional or natural proprieties of people [6, 29]. Instead, they should be seen as the result of active social processes, in which language holds socio-political connotations, which lead to the categorisation of individuals [30]. In research, *identity* (understood as an internal feeling of oneself) is a concept that should be made visible in data collection methods, to consider the population's diversity and their health outcomes.

To conclude, non-binary people or people who did not identify with the man/woman binomial presented high proportions of anxiety and depression during lockdown, compared to binary people. However, there are differences between binary people. People who identify as male have better mental health outcomes than non-binary people. While no differences were observed between non-binary people and women. This highlights the differences in social and economic privileges that men have over other gender identities. In fact, currently should be study how the pandemic impact exacerbated pre-existing social inequities, such as adverse social, employment and economic conditions in some population. This suggests the need to conduct studies using different methodological designs, to make pertinent recommendations. We suggest increasing the evidence on the diversity in the sex-gender system, to deepen our understanding on its relationship with health status and socio-structural determinants of health. Cultural diversity and racism should also be considered. In addition, interventions focused on challenging gender stereotypes should be promoted to reduce social inequities of health.

## Supporting information

**S1 Table. Mental health, sociodemographic characteristics and social and health-related factors among people who do not identify within the man/woman binomial during**

**COVID-19 lockdown in Spain.**
(DOCX)

**S2 Table. Mental health, sociodemographic characteristics and social and health-related factors among people who do not identify within the man/woman binomial and men and women during COVID-19 lockdown in Spain.**
(DOCX)

**S3 Table. Associations between sociodemographic characteristics, gender identity, social and health-related factors, and mental health during COVID-19 lockdown in Spain.**
(DOCX)

**S4 Table. GAD-7[1] items.**
(DOCX)

**S5 Table. PHQ-9[1] items.**
(DOCX)

**S6 Table. GAD-7[1] items.**
(DOCX)

**S7 Table. PHQ-9[1] items.**
(DOCX)

## Acknowledgments

We would like to thank Talita Duarte-Salles for her contributions to this manuscript and Patryck Bialoskorski for his contributions revising the language in this article. In addition, we would like to thank all reviewers for their comments, which have helped improving the quality of this article.

## Author Contributions

**Conceptualization:** Constanza Jacques-Aviñó, Tomàs López-Jiménez, Laura Medina-Perucha.

**Data curation:** Tomàs López-Jiménez.

**Formal analysis:** Tomàs López-Jiménez.

**Funding acquisition:** Anna Berenguera.

**Investigation:** Constanza Jacques-Aviñó, Laura Medina-Perucha, Jeroen de Bont.

**Methodology:** Constanza Jacques-Aviñó, Tomàs López-Jiménez, Laura Medina-Perucha, Jeroen de Bont, Anna Berenguera.

**Resources:** Tomàs López-Jiménez, Anna Berenguera.

**Validation:** Constanza Jacques-Aviñó, Laura Medina-Perucha, Jeroen de Bont, Anna Berenguera.

**Writing – original draft:** Constanza Jacques-Aviñó.

**Writing – review & editing:** Constanza Jacques-Aviñó, Tomàs López-Jiménez, Laura Medina-Perucha, Jeroen de Bont, Anna Berenguera.

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
