## [Decision Letter · Decision Letter 0]

18 Feb 2021

PONE-D-20-29954

Social conditions and mental health during COVID-19 lockdown among people who do not identify with the man/woman binomial in Spain

PLOS ONE

Dear Dr. Jacques Aviñó,

Thank you for submitting your manuscript to PLOS ONE. After careful consideration, we feel that it has merit but does not fully meet PLOS ONE’s publication criteria as it currently stands. Therefore, we invite you to submit a revised version of the manuscript that addresses the points raised during the review process.

We look forward to receiving your revised manuscript.

Kind regards,

Sandra Carvalho

Academic Editor

PLOS ONE

Journal Requirements:

Reviewers' comments:

Reviewer's Responses to Questions

**Comments to the Author**

1. Is the manuscript technically sound, and do the data support the conclusions?

Reviewer #1: Partly

Reviewer #2: Partly

2. Has the statistical analysis been performed appropriately and rigorously? 

Reviewer #1: No

Reviewer #2: No

3. Have the authors made all data underlying the findings in their manuscript fully available?

Reviewer #1: No

Reviewer #2: Yes

4. Is the manuscript presented in an intelligible fashion and written in standard English?

Reviewer #1: Yes

Reviewer #2: Yes

5. Review Comments to the Author

Reviewer #1: Dear editor,

Thank you for the opportunity to review the manuscript “Social conditions and mental health during COVID-19 lockdown among people who do not identify with the man/woman binomial in Spain”. It explores, as the title suggests, the social and health-related factors associated with mental health among people who do not identify with the man/woman binomial during COVID-19 lockdown. In the end, the authors conclude that people who did not identify with the man/woman binomial presented high proportions anxiety and depression during lockdown when compared to binary people. Although the manuscript brings to light an interesting and, commonly, neglected field of research, I do have two major concerns. Please, find them below:

1. Concerning the logistic models, I might have some suggestions:

a) The description of the models could improve. Precisely, the authors could make a bit more clear that the dependent variables were moderate/severe for anxiety and depression.

b) Sorry if I missed the information in the text but, why did the authors include 71 participants in the model if they had 72 participants initially? All the 72 participants fully complete the survey?

c) Another interesting analysis that can be easily done and might add some important topics in the discussion is to randomly select a matched sample for sociodemographic characteristics of 72 participants from the main database (n=7125) and (i) compare both groups regarding all mental health, and social and health-related factors. By doing so, it would be also possible to (ii) run the analysis in another way around, which means including being someone who does not identify with the man/woman binomial as an independent variable for moderate/severe anxiety and depression. Note that, although this might extrapolate the main aim of the study, it will fulfill a huge existing gap: How does the authors know if there is any specificity in this sample that should be taken into account and that it differs from the general population? In other words, how did the authors infer that it could have exacerbated pre-existing social inequities, such as adverse social, employment and economic conditions? For instance, by doing so, the authors would not need to hypothesized that “this could be explained due to the discrimination that non-binary people experience whenseeking employment”, but actually prove it. Additionally, how did the authors conclude that people who did not identify with the man/woman binomial presented high proportions of anxiety and depression during lockdown when compared to binary people, if the authors did not perform any group comparison, even being possible to do so? In my point of view, this is the most weak point of the present manuscript, because (a) it could have been done and (b) it would considerably increase the possible conclusions about the data. For instance, showing that not people who did not identify themselves with the man/woman binomial is a risk-factor, per se, independently of other variables; showing that even when compared to a matched sample people who did not identify themselves with the man/woman binomial indeed have more symptoms, etc. etc. In sum, the data analysis could have been far more complete.

d) Was any specific finding concerning the different items in the GAD7 or PHQ9? I am asking because another interesting finding could be to identify questionnaire items where people who do not identify with the man/woman binomial tend to report as more severe. Especially, if the authors are planning to use this data in qualitative future study, as they report. For instance, the authors could find that people who do not identify with the man/woman binomial report more self injuring thoughts. Again, another easy finding that could be extracted from the data and could considerably enrich the quality of the manuscript.

e) Which software was used for data analysis?

f) How does this information “All members of the research team are highly sensitive towards the different axes of social inequities, and the multiple power structures that impact the population unequally” adds something for the study? I mean, as a human science researcher, isn't it redundant to randomly state this in the method?

2. Concerning the survey:

a) Which software was used to run the online survey? How was the advertisement done? The authors provide any information about it. Was it through electronic media and a snowball sampling method, whereby the researcher invited the participants to share the survey with their contacts? There is far too little information on this regard.

b) How did the authors check for the reliability of the participants' answers? In online or extensive surveys, it might be useful to include some methods to decrease the odds of considering in the analysis someone that answered it randomly. For instance, including two similar questions in different moments to confirm if the answers will be also similar (sometimes it can be done by checking similar questions between and within questionnaires). Another strategy could be to check for mistakes or weird answers in basic questions such as age. Moreover, did the authors include a question asking whether the participant had already answered the survey? How much time, on average was necessary to complete the survey? Sometimes, this could also be used as a criteria to exclude non-reliable answers. For instance, if someone took much less than the mean time and its variance, it could indicate non-reliable answers.

3. Concerning the results and discussion:

a) In general, I think that both are adequate for the implemented method. Nevertheless, if the authors intend to address some of the major methodological concerns and considerably increase the quality of the study and the possible conclusions on the observed behaviour it will change both sections.

b) I would recommend reporting the results from Table 2 as a figure. It is a minor suggestion, but I think it turns more interesting for the reader. Don’t you?

c) The authors discussed that “this study contributes to the debate on the categorisation of sex-gender identities in health surveys and other registries, to question and challenge normative ways of classification”, however, they did not provide some options about how this could have been done or how to improve it in future studies. This is crucial, especially because the authors indeed add that “The research team will consider which could be the best way to frame questions around sex/gender for future surveys”. I mean, the main science goal is to learn from our own and others mistakes, right? How can I improve my research practice and reframe this question about gender/sex?

d) I did not find the fully available data.

Once more, thank you for the opportunity,

Best regards,

Reviewer #2: Line 51- Why is queer italicized?

Lines 65, 75-76- The terms sex-gender/sexual-gender is unclear. Please replace or define

Since sexual identity is generally used to refer to the sexual orientation that one identifies with, I don’t think this is a good term to use- I encourage you to find an alternative; I find the description of why you are not using the term gender identity unclear, please expand.

Line 116- Death of loved one

Lines 177-179- Without doing a significance test, I don’t think you can make the claim that non-binary people’s experiences are worse than men or women’s

I encourage another round of proofreading/copy editing

There is a lot of important work here, and I am so glad you are considering a vulnerable and under-researched population. I have concerns that your current methods do not allow you to say that non-binary people are experiencing greater mental health problems than their binary counterparts or that COVID is exacerbating these inequities. I would argue that associations between unemployment pre-COVID and depression/anxiety may suggest COVID is not exacerbating these symptoms but rather this problem was pre-existing. I encourage using a comparative analysis- perhaps an interaction of gender identity/predictor variable and mental health outcomes- using only variable that have shifted during COVID OR changing the framing to highlight particularly vulnerable subgroups of non-binary folks.

6. PLOS authors have the option to publish the peer review history of their article (what does this mean?). If published, this will include your full peer review and any attached files.

Reviewer #1: No

Reviewer #2: **Yes: **Briana L McGeough

---

## [Author Response · Author response to Decision Letter 0]

21 Apr 2021

Dear Editor,

Please find enclosed our revised manuscript, entitled “Social conditions and mental health during COVID-19 lockdown among people who do not identify with the man/woman binomial in Spain” by Constanza Jacques-Aviñó, Tomás López-Jimenez, Laura Medina-Perucha, Jeroen de Bont and Anna Berenguera for publication consideration in the Plos One Journal.

We want to thank both reviewers for their comments. We have positively valued all the feedback. It has helped us to reflect and considerably improve the quality of our manuscript. We have added a special mention of the reviewers in the acknowledgement section. 

We answered all the reviewers’ comments and modified the manuscript accordingly. Pages and lines refer to a marked-up copy version.

1. Concerning the logistic models, I might have some suggestions:

a) The description of the models could improve. Precisely, the authors could make a bit more clear that the dependent variables were moderate/severe for anxiety and depression.

We appreciate the reviewer's comment. We have added this information accordingly in method section:

- Anxiety was considered as moderate and severe level (page 5, line 121-122) 

- Depression was considered as moderate and moderately-severe/severe level. (page 6, line 124-125)

b) Sorry if I missed the information in the text but, why did the authors include 71 participants in the model if they had 72 participants initially? All the 72 participants fully complete the survey?

We appreciate the reviewer's comment. One case had a missing value. After discussion with the research team, we decided to impute this value and now we have included 72 persons in the models. We added actualized table 1 (page 8 and 9, line 170-173) and table 2 (page 11, line 193 -200) accordingly.

We have added this information accordingly in the method section (page 6, line 139-142).

c) Another interesting analysis that can be easily done and might add some important topics in the discussion is to randomly select a matched sample for sociodemographic characteristics of 72 participants from the main database (n=7125) and (i) compare both groups regarding all mental health, and social and health-related factors. By doing so, it would be also possible to (ii) run the analysis in another way around, which means including being someone who does not identify with the man/woman binomial as an independent variable for moderate/severe anxiety and depression. Note that, although this might extrapolate the main aim of the study, it will fulfill a huge existing gap: How does the authors know if there is any specificity in this sample that should be taken into account and that it differs from the general population? In other words, how did the authors infer that it could have exacerbated pre-existing social inequities, such as adverse social, employment and economic conditions? For instance, by doing so, the authors would not need to hypothesized that “this could be explained due to the discrimination that non-binary people experience when seeking employment”, but actually prove it. Additionally, how did the authors conclude that people who did not identify with the man/woman binomial presented high proportions of anxiety and depression during lockdown when compared to binary people, if the authors did not perform any group comparison, even being possible to do so? In my point of view, this is the most weak point of the present manuscript, because

(a) it could have been done and (b) it would considerably increase the possible conclusions about the data. For instance, showing that not people who did not identify themselves with the man/woman binomial is a risk-factor, per se, independently of other variables; showing that even when compared to a matched sample people who did not identify themselves with the man/woman binomial indeed have more symptoms, etc. etc. In sum, the data analysis could have been far more complete.

We very much appreciate this suggestion and have carried out the analysis as suggested by the reviewer. We performed the match with binary people, choosing 4 binary people for each non-binary person. Their characteristics were as follows: same proportion of sex (2 men and 2 women), same age (±1 year) and level of education.

Our results suggest statistically significant difference in anxiety levels between non-binary people and binary people (41.7% vs 27.1%). Regarding depression symptoms non-binary had 30.6% compared to 28.8% in binary people but this difference was not statistically significant.

However, we went a step further, since we know that there are differences in mental health between men and women, we stratified by gender. Our results show statistically significant differences between anxiety levels between non-binary people (41.7%) and men (16.0%). Regarding women we did not find statistically significant differences (38.2% vs. 41.7%) for anxiety and was did not find differences between depression in non-binary and women (38.9% vs 30.6%).

We have added this information accordingly in the method section (page 6, line 133-136) and we have added this information accordingly in the results section (page 11-12, line 214-225).

Furthermore, when we introduced the non-binary people as an independent variable and the matched binary people, we could observe that identifying as non-binary person was a risk factor for anxiety (ORa: 1.89 CI: 1.05-3.39) but not for depression (ORa: 0.93; CI: 0.50-1.74). Therefore, we have added this information accordingly in the results section (page 12, line 222-225).

More information is available in supplementary material.

d) Was any specific finding concerning the different items in the GAD7 or PHQ9? I am asking because another interesting finding could be to identify questionnaire items where people who do not identify with the man/woman binomial tend to report as more severe. Especially, if the authors are planning to use this data in qualitative future study, as they report. For instance, the authors could find that people who do not identify with the man/woman binomial report more self injuring thoughts. Again, another easy finding that could be extracted from the data and could considerably enrich the quality of the manuscript.

We appreciate this comment and we analysed each item of GAD-7 and PHQ-9 individually. Our results show differences between binary and non-binary people in only one item (“Being so restless that it is hard to sit still”) of GAD-7. Therefore, we decided stratified by gender to find out deeper if there were differences between genders. We found statistical significant differences between non-binary people and men in following GAD-7 items: “worrying too much about different things” and “having trouble relaxing and being so restless that it is hard to sit still”. On other hand, we found statistically significant differences between non binary people and men in following PHQ-9 items: “feeling down, depressed, or hopeless” and “trouble falling or staying asleep or sleeping too much and feeling tired or having little energy”.

We have added this information accordingly in the results section (page 12, line 227- 

232).

e) Which software was used for data analysis?

We used STATA 15.1; we specified this in the method section (page 6, line 146).

f) How does this information “All members of the research team are highly sensitive towards the different axes of social inequities, and the multiple power structures that impact the population unequally” adds something for the study? I mean, as a human science researcher, isn't it redundant to randomly state this in the method?

This is a good point and we appreciate this comment. We have thoroughly discussed this sentence and we think it is important to address it in our manuscript based on two arguments. First, our team believes that the research position should be explicit. It is very important in order to be transparent in how and why we want to conduct this study. Second, we know that there is an ongoing discussion on non-binary and trans* people from different approaches. For some people these groups are not considered worthy of recognition. Nevertheless, we believe that they should be seen and included in research, since giving them visibility is a way to fight against social inequities in health.

2. Concerning the survey:

a) Which software was used to run the online survey? How was the advertisement done? The authors provide any information about it. Was it through electronic media and a snowball sampling method, whereby the researcher invited the participants to share the survey with their contacts? There is far too little information on this regard.

We thank the reviewer for this comment and we included more information about the online survey, electronic media and recruitment in the methods section (page 4, line 88-94).

b) How did the authors check for the reliability of the participants' answers? In online or extensive surveys, it might be useful to include some methods to decrease the odds of considering in the analysis someone that answered it randomly. For instance, including two similar questions in different moments to confirm if the answers will be also similar (sometimes it can be done by checking similar questions between and within questionnaires). Another strategy could be to check for mistakes or weird answers in basic questions such as age. Moreover, did the authors include a question asking whether the participant had already answered the survey? How much time, on average was necessary to complete the survey? Sometimes, this could also be used as a criteria to exclude non-reliable answers. For instance, if someone took much less than the mean time and its variance, it could indicate non-reliable answers.

The survey was created based on the research team's background and expertise. We analysed if there were any incoherent answers and did not find any. The average time to answer the survey was 10 minutes; this information was explained on the front page of the survey and we included it in the manuscript in the method section (page 4-5, line 95-98).

Regarding asking two similar questions at different times, we did not do it, nor we could limit whether someone could participate more than once. For the purposes of this study, we conducted a review of survey entry to assess internal consistency, for example, comparing age and education level. We have added this limitation of the study in the discussion section (page 15, line 305-307).

3. Concerning the results and discussion:

a) In general, I think that both are adequate for the implemented method. Nevertheless, if the authors intend to address some of the major methodological concerns and considerably increase the quality of the study and the possible conclusions on the observed behaviour it will change both sections.

We fully agree with the reviewer's comment, therefore we have added more information in the results section regarding the previous comment and included some additional sentences in discussion section (page 12-13, line 243-249). 

We also have added this information accordingly in the discussion section (page 16 line 329 and line 333, 334, 339).

b) I would recommend reporting the results from Table 2 as a figure. It is a minor suggestion, but I think it turns more interesting for the reader. Don’t you?

We appreciate the reviewer's suggestion but since the confidence intervals are very wide, we rather think it is not optimal to report the results in a figure.

c) The authors discussed that “this study contributes to the debate on the categorisation of sex-gender identities in health surveys and other registries, to question and challenge normative ways of classification”, however, they did not provide some options about how this could have been done or how to improve it in future studies. This is crucial, especially because the authors indeed add that “The research team will consider which could be the best way to frame questions around sex/gender for future surveys”. I mean, the main science goal is to learn from our own and others mistakes, right? How can I improve my research practice and reframe this question about gender/sex?

We appreciate the reviewer's comment, as the team has made a thorough review of this issue in the current survey (our second cross-sectional study). We have added this information accordingly in discussion section (page 14, line 295-296 and page 15, line 299-302).

d) I did not find the fully available data.

We apologize for this, but the Ethical Committee does not allow us to share the data publicly as our data contain sensitive personal information and cannot be fully anonymised.

Reviewer #2: Line 51- Why is queer italicized?

We appreciate this comment and we have made changes accordingly.

Lines 65, 75-76- The terms sex-gender/sexual-gender is unclear. Please replace or define Since sexual identity is generally used to refer to the sexual orientation that one identifies with, I don’t think this is a good term to use- I encourage you to find an alternative; I find the description of why you are not using the term gender identity unclear, please expand.

We are grateful for the reviewer's comment, as the research team has discussed this issue several times. We have considered relevant to address gender identity or sexual identity indistinctly in the manuscript. For the analyses, the sex-gender system is intrinsic to the term "identity". That is, we do not differentiate between sexual identity and gender identity. We assume that respondents understood the question and chose the response categories that identified them (Women/Man/Non-binary/I do not identify). We have added this information accordingly in method section (page 5, line 109-116).

However, after this reflexion we have considered using the term gender identity for the second survey study (mentioned previously), which is currently ongoing. In addition, we have included other questions that may delve deeper into issues related to the sex-gender system. We have included this in the discussion section of the manuscript (page 15, line 299-302).

Line 116- Death of loved one

We appreciate this comment and we corrected this misspelling.

Lines 177-179- Without doing a significance test, I don’t think you can make the claim that non-binary people’s experiences are worse than men or women’s

I encourage another round of proofreading/copy editing

There is a lot of important work here, and I am so glad you are considering a vulnerable and under-researched population. I have concerns that your current methods do not allow you to say that non-binary people are experiencing greater mental health problems than their binary counterparts or that COVID is exacerbating these inequities. I would argue that associations between unemployment pre-COVID and depression/anxiety may suggest COVID is not exacerbating these symptoms but rather this problem was pre-existing. I encourage using a comparative analysis- perhaps an interaction of gender identity/predictor variable and mental health outcomes- using only variable that have shifted during COVID OR changing the framing to highlight particularly vulnerable subgroups of non-binary folks.

We appreciate this suggestion and, therefore, we have carried out the relevant analyses. We have performed matching with binary persons (2 men and 2 women for each non-binary person), who were of the similar age (+- 1 year) and the same education level for comparison. We have added this information accordingly in the methods section (page 6, line 133-135) and in the results section (page11-12, line 214-225). The results of the analyses are available in the supplementary materials.

---

## [Decision Letter · Decision Letter 1]

4 Aug 2021

Social conditions and mental health during COVID-19 lockdown among people who do not identify with the man/woman binomial in Spain

PONE-D-20-29954R1

Dear Dr. Avino,

We’re pleased to inform you that your manuscript has been judged scientifically suitable for publication and will be formally accepted for publication once it meets all outstanding technical requirements.

Kind regards,

Gerard Hutchinson, MD

Academic Editor

PLOS ONE

Additional Editor Comments (optional):

Reviewers' comments:

Reviewer's Responses to Questions

**Comments to the Author**

1. If the authors have adequately addressed your comments raised in a previous round of review and you feel that this manuscript is now acceptable for publication, you may indicate that here to bypass the “Comments to the Author” section, enter your conflict of interest statement in the “Confidential to Editor” section, and submit your "Accept" recommendation.

Reviewer #1: All comments have been addressed

2. Is the manuscript technically sound, and do the data support the conclusions?

Reviewer #1: Yes

3. Has the statistical analysis been performed appropriately and rigorously? 

Reviewer #1: Yes

4. Have the authors made all data underlying the findings in their manuscript fully available?

Reviewer #1: No

5. Is the manuscript presented in an intelligible fashion and written in standard English?

Reviewer #1: Yes

6. Review Comments to the Author

Reviewer #1: Dear Editor,

Thank you once more for the opportunity to review the manuscript “Social conditions and mental health during COVID-19 lockdown among people who do not identify with the man/woman binomial in Spain. The authors fully attended to all my concerns and did a great work by improving the data analysis, which had a substantial impact in the results and, consequently, to the overall picture and possible interpretations. The method was also improved concerning the data collection strategy, which was a major point before.

As a minor point, I would suggest the authors to say a word about “We analysed if there were any incoherent answers and did not find any”. I mean, how? Once more, this is not crucial.

Best regards,

7. PLOS authors have the option to publish the peer review history of their article (what does this mean?). If published, this will include your full peer review and any attached files.

Reviewer #1: **Yes: **Bruno Kluwe-Schiavon

---

## [Editor Report · Acceptance letter]

13 Aug 2021

PONE-D-20-29954R1 

Social conditions and mental health during COVID-19 lockdown among people who do not identify with the man/woman binomial in Spain 

Dear Dr. Jacques-Aviñó:

I'm pleased to inform you that your manuscript has been deemed suitable for publication in PLOS ONE. Congratulations! Your manuscript is now with our production department. 

Kind regards, 

on behalf of

Dr. Gerard Hutchinson 

Academic Editor

PLOS ONE